# Ciliary muscles contraction leads to axial length extension—The possible initiating factor for myopia

**Zhao-Yang Meng[1], Lin Yang[2], Peng Zhou[ID][3,4] ***

**1** Department of Ophthalmology, Beijing Friendship Hospital, Capital Medical University, Beijing, China,
**2** Department of Ophthalmology, Visionly Plus Eye Hospital, Beijing, China, **3** Department of Ophthalmology,
Parkway Gleneagles Medical and Surgical Center, Shanghai, China, **4** Department of Ophthalmology,
Parkway Hong Qiao Medical Center, Shanghai, China

* drzhoupeng@gmail.com

## Abstract

### Purpose

This study aimed to investigate the underlying factors driving the onset of myopia, specifically the role of the ciliary muscle's contraction in the elongation of the axial length of the eye.

### Methods

The retrospective study was conducted utilizing data from three ophthalmic centers in Shanghai and Beijing. Both Chinese and Caucasian children were involved. The axial length of the subjects' eyes was measured in both relaxed and contracted state of the ciliary muscle. A comprehensive mechanical model was also developed to observe the influence of ciliary muscle contraction on the axial length.

### Results

This study included a sample of 198 right eyes of 198 myopic children. Of these, 97 were male and 101 were female, 126 were of Chinese ethnicity and 72 were Caucasian. The age of onset for myopia ranged from 5.9 to 16.9 years old. The axial length of the eye decreased $0.028 \pm 0.007$mm following dilation, indicating relaxation of the ciliary muscle ($t_{paired\ student} = 15.16$, $p = 6.72 \times 10^{-35}$). In contrast, ciliary muscle contraction resulted in an increase in axial length. Considering proportionality, a significant 90.4% (179 eyes) exhibited a reduced axial length, while a minor 9.6% (19 eyes) demonstrated an increase post-mydriasis. Finite element modeling demonstrated that muscle contraction caused a tension force that transmits towards the posterior pole of the eye, causing it to extend posteriorly.

### Conclusion

The contraction of the ciliary muscle leads to an extension of the axial length. This could potentially be the initiating factor for myopia.

**Data Availability Statement:** All relevant data are within the manuscript and its Supporting Information files.

**Funding:** The author(s) received no specific funding for this work.

**Competing interests:** The authors have declared that no competing interests exist.

# Introduction

Myopia is recognized as a substantial global public health issue [1]. In East and Southeast Asia, the prevalence of myopia among young adults in middle-school age ranges between 70% and 90% [2, 3]. A systematic review and meta-analysis indicated that the global prevalence of myopia was 22.9%, with 2.7% experiencing high myopia in 2020. Projections suggest a significant escalation by 2050, with an estimated 4758 million individuals (49.8%) likely to be affected by myopia and 938 million (9.8%) by high myopia [4]. High myopia carries the risk of various complications, including cataracts, choroidal neovascularization, retinal detachment, glaucoma, and macular atrophy, which can have a significant impact on visual impairment and potentially lead to irreversible blindness [5, 6]. Therefore, it is imperative to implement measures to prevent the progression of myopia.

Both genetic and environmental factors are recognized as influential factors in the development of myopia [7]. The most significant mechanism involved in the development of myopia is the elongation of the axial length of the eyeball [8, 9]. Among the environmental contributors to myopia, factors such as extensive engagement in near work, limited participation in outdoor activities, and disturbances in circadian rhythm control have been identified [10].

Previous investigations into the etiology of myopia have revealed several potential factors and mechanisms contributing to its development. Accommodation, the eye's ability to adjust focus, has been extensively studied in relation to myopia. While certain studies have identified associations between accommodative lag (a delay in focusing) and myopia, others propose that accommodative lag is an outcome rather than a causative factor in myopia [11, 12]. Prior investigations have noted the correlation between ciliary muscle contraction and myopia [13]. Multi-factorial analyses of ocular accommodation, which take into account the morphology of the ciliary muscle, the depth of the anterior chamber, as well as the evaluation of nutritional intake and metabolic indicators, may illuminate the pathogenesis and contribute to the development of innovative management strategies for myopia [13]. Nevertheless, these studies have not delved into the mechanisms by which contractions of the ciliary muscle lead to the elongation of the eye's axial length. Another area of interest resides in the anatomy and biomechanics of the ciliary body, which governs lens shape. Specifically, myopic individuals exhibit a larger and stiffer ciliary muscle, potentially impacting accommodative function [14]. The concept of peripheral defocus, characterized by hyperopic defocus in the outer retinal regions, has emerged as a possible contributing element in myopia development [15]. Nevertheless, the role of peripheral defocus as either a cause or consequence of myopia remains an ongoing debate [16]. Moreover, various signaling pathways, including dopamine, nitric oxide, TGF-β, HIF-1α, among others, are crucial in the occurrence and development of myopia [17]. However; previous studies did not established a direct link between these potential factors and the elongation of the axial length of the eyeball. The possible initiating factor for the elongation of the axial length in myopia remains unclear.

In order to elucidate the underlying factors driving the onset of myopia, it is imperative to adhere to the "First Principles" and delve into the etiology of this condition. A First Principles is a basic assumption that cannot be deduced any further. It is widely acknowledged that excessive engagement in near work activities serves as the primary catalyst for the development of myopia [16]. Therefore, it is crucial to investigate the alterations occurring in ocular structures when focusing on objects in close proximity. Specifically, during near work tasks, the contraction of the ciliary muscle within the eye takes place. Previous studies have hypothesized that this muscle contraction may induce various forms of optical defocus, ultimately culminating in the elongation of the eyeball via intricate mechanisms. However, taking a step back to the first principles, it becomes essential to question whether the sole contraction of the ciliary

muscle can indeed result in the morphological transformation of the spherical eyeball into an elliptical shape, thereby instigating elongation. In light of this comprehensive analysis, we have devised an experiment that adheres to this line of inquiry.

This retrospective study aimed to compare the axial length of the eye in two distinct states: relaxed state of the ciliary muscle (after dilation) and contracted state (prior to dilation). Additionally, a comprehensive mechanical model was devised to examine the transmission of forces through the eye wall during ciliary muscle contraction and its subsequent influence on the axial length. The main objective of this investigation is to elucidate the underlying factors that contribute to the onset of myopia.

## Material and methods

### Study design and subjects

This retrospective study was conducted at ophthalmic centers, namely Parkway Gleneagles Medical and Surgical Center (PG), Shenton Health Hong Qiao Medical Center (SH), and Visionly Plus Eye Hospital (VP). PG and SH are internationally renowned medical facilities located in Shanghai, serving both Chinese and Caucasian patients. Meanwhile, VP exclusively caters to Chinese patients.

Data collection spanned from January 1st, 2022 to June 30th, 2023. The data were accessed for research purposes on September 1st, 2023. The inclusion criteria for our subjects were as follows: (i) aged between 5 and 18 years, (ii) cycloplegic spherical equivalent (SE) of the worst eye less than −0.5 diopters, (iii) absence of any significant ocular diseases, and (iv) informed consent provided by parents or guardians. (v) No intervention (for example, orthokeratology or low-concentration atropine eye drops) was used.

### Ethics statement

This study followed the principles outlined in the Declaration of Helsinki. The study protocol was approved by the Ethics Committee and Institutional Review Board (IRB) of Parkway Gleneagles Medical and Surgical Center (No. 202302). The IRB waived the informed consents for this retrospective anonymous study.

### Examination of axial length

"The methodology for axial length measurement is consistent with that used in our previous studies [18]. Briefly, the methodology employed in this study was standardized across all three ophthalmic centers, ensuring consistency among the ophthalmologists. Slit lamp examination with the aid of a slit lamp lens (Digital Wide Field, Volk, USA) was conducted by trained ophthalmologists to assess both the anterior and posterior segments of the eyes.

The measurement of axial length was performed twice: first before dilation (in the contracted state of the ciliary muscle) and then after dilation (in the relaxed state of the ciliary muscle). The AL-Scan Optical Biometer (Nidek, Japan), an optical biometry device, was utilized to measure the axial length before dilation. Following the administration of three drops of Mydrin P (Tropicamide 0.5%, phenylephrine HCl 0.5%; Santen Pharmaceutical, Shiga, Japan) at 5-minute intervals, and a subsequent waiting period of 30 minutes for full cycloplegia, the same optical biometry device was used to measure the axial length once more. Axial length measurements were obtained for both eyes, but for the purposes of this study, data from the right eye of each subject were utilized.

## Biomechanical simulation

The construction of the Finite Element Model is analogous to that employed in previous studies [19]. In brief, a 2-D finite element biomechanical model of the eye was developed using the ABAQUS 2020 software (Dassault Systèmes SIMULIA Corp.). As the focus of this study does not involve alterations in the lens, we integrated the ciliary muscle with the lens to create a "ciliary muscle-lens complex". The simulated parameters for the experimental conditions were as follows: an axial length of the eye measuring 22.78mm, a distance of 5.73mm from the cornea to the posterior surface of the lens, a distance of 17.05mm from the posterior surface of the lens to the posterior pole of the eye, and a thickness of the eye wall of 0.67mm. The source file of this finite element biomechanical model is in the supplementary data (S1 File and S1 Fig) of this manuscript, which can be opened with the ABAQUS software.

## Statistical analysis

For the calculation of the required sample size for this study, we employed an online Sample Size Calculator (https://homepage.univie.ac.at/robin.ristl/samplesize.php). The determination of the sample size was informed by preliminary pilot data and guided by relevant literature. The statistical method selected was the paired t-test, with the mean difference set at 0.01 and the standard deviation of differences at 0.03. We established the significance level (alpha) for a two-tailed test at 0.01, and the desired statistical power at 0.9. These parameters yielded a required sample size of 138 pairs. In the present study, a total of 198 eyes were included, fulfilling the criteria for the requisite sample size.

The data were analyzed using the R programming language (version 4.1.3). Paired student t-test was performed to compare the axial length before dilation (in the contracted state of the ciliary muscle) and after dilation (in the relaxed state of the ciliary muscle). A P-P (Probability-Probability) plot is utilized to ascertain the conformity of a data set with a normal distribution. The Pearson correlation coefficient is employed to assess whether there are differences in the data between both eyes. The significance level was set at $p < 0.01$.

We employed an online tool to compute the effect size for a before-and-after study (Paired T-test) (https://sample-size.net/effect-size-study-paired-t-test/). The significance level (alpha) for a two-tailed test was set at 0.01, and the desired statistical power at 0.9, with a sample size of 198. Based on the results from a pilot study, the standard deviation of the change in the outcome was established at 0.01627. Subsequent calculations yielded an effect size of 0.004.

# Results

## Study sample characteristics

This study included a sample of 198 myopic children with 198 right eyes. Of these, 97 (48.99%) were male and 101 (51.01%) were female. Among the participants, 126 (63.64%) were of Chinese ethnicity and 72 (36.36%) were Caucasian. The age ranged from 5.9 to 16.9 years old, with a median age of 9.8 years old. The mean age, gender distribution, race, spherical equivalent (SE), axial length (AL) before and after dilation are summarized in Table 1. The mean spherical equivalent was -2.44 ±0.21 diopter sphere (Ds, ranging from -0.50 Ds to -6.00 Ds), and the mean cylinder value was -1.01 ±0.26 diopter of cylinder (Dc)(ranging from -0.00 Dc to -2.50 Dc). Anisometropia, defined as a refractive error difference of 1.00D or more between the eyes, was observed in six children (3.03%).

### The change of axial length before and after mydriasis

Adiscernible shortening in the eye's axial length was found following mydriasis when compared to before. More specifically, contraction of the ciliary muscle results in an increase in axial length, while relaxation leads to a reduction.

Fig 1A illustrates the changes in axial length before and after the process of mydriasis. The mean axial length before mydriasis was found to be 24.71 ± 0.23mm. Conversely, it was observed to be 24.68 ± 0.22mm after mydriasis, indicating a decrease of 0.028 ± 0.007mm compared to its initial measurement. The variation in axial length before and after mydriasis was determined to be statistically significant ($t_{paired\ student}$ = 15.16, p = 6.72 x $10^{-35}$). Across both eyes, no apparent axial length differences were identified (Pearson correlation coefficient before dilation r(196) = 0.9636; P<0.001, after dilation r(196) = 0.9507, P<0.001).

Considering proportionality, a significant 90.4% (179 out of 198 eyes) exhibited a reduced axial length, while a minor 9.6% (19 out of 198 eyes) demonstrated an increase post-mydriasis (as depicted in Fig 1B).

### Force distribution of ciliary muscle contraction on the eye ball

We constructed a finite element model of the eyeball using ABAQUS software. As this research does not involve changes in the shape of the lens, we have simplified the ciliary muscle and lens into a "ciliary muscle-lens complex". The thickness of each part of the eye is set according to a "standard eye". The ABAQUS model database of ciliary muscles contraction is in the Supporting Information 1 of this manuscript.

Fig 2A demonstrates that upon the contraction of the ciliary muscle, the tension (depicted in red, yellow, and green) starts from the point of attachment of the ciliary muscle and transmits towards the back of the eye, eventually reaching the posterior pole. Interestingly, there is little tension transmission to the cornea, with most being transmitted towards the posterior pole. Fig 2B illustrates that when the ciliary muscles contract, the axial length of the eye extends posteriorly. The animation of tension changes during ciliary muscle contraction is in the Supporting Information 2 of this manuscript.

## Discussion

The research involved a sample size of 198 myopic children who were of both Chinese and Caucasian descent. Significant changes in axial length were observed pre and post-mydriasis,

**Table 1. Demographic and biometric at baselines.**

| | | | |
|---|---|---|---|
| **N** | 198 | | |
| **Age** | 10.32 ± 2.30 | | |
| **Gender** | | | |
| • Male | 97 (48.99%) | | |
| • Female | 101 (51.01%) | | |
| **Race** | | | |
| • Chinese | 126 (63.64%) | | |
| • Caucasian | 72 (36.36%) | | |
| | **Before dilation** | **After Dilation** | **P value** |
| **Spherical Equivalent (D)** | -2.44 ± 0.21 | -2.03 ± 0.23 | 1.76 x $10^{-9}$ |
| • PCC between right eye and left eye | 0.97 (P <0.001) | 0.98 (P <0.001) | |
| **Axial Length (mm)** | 24.71 ± 0.23 | 24.68 ± 0.22 | 6.72 x $10^{-35}$ |
| • PCC between right eye and left eye | 0.96 (P <0.001) | 0.95 (P <0.001) | |

D: diopter; PCC: Pearson correlation coefficient.

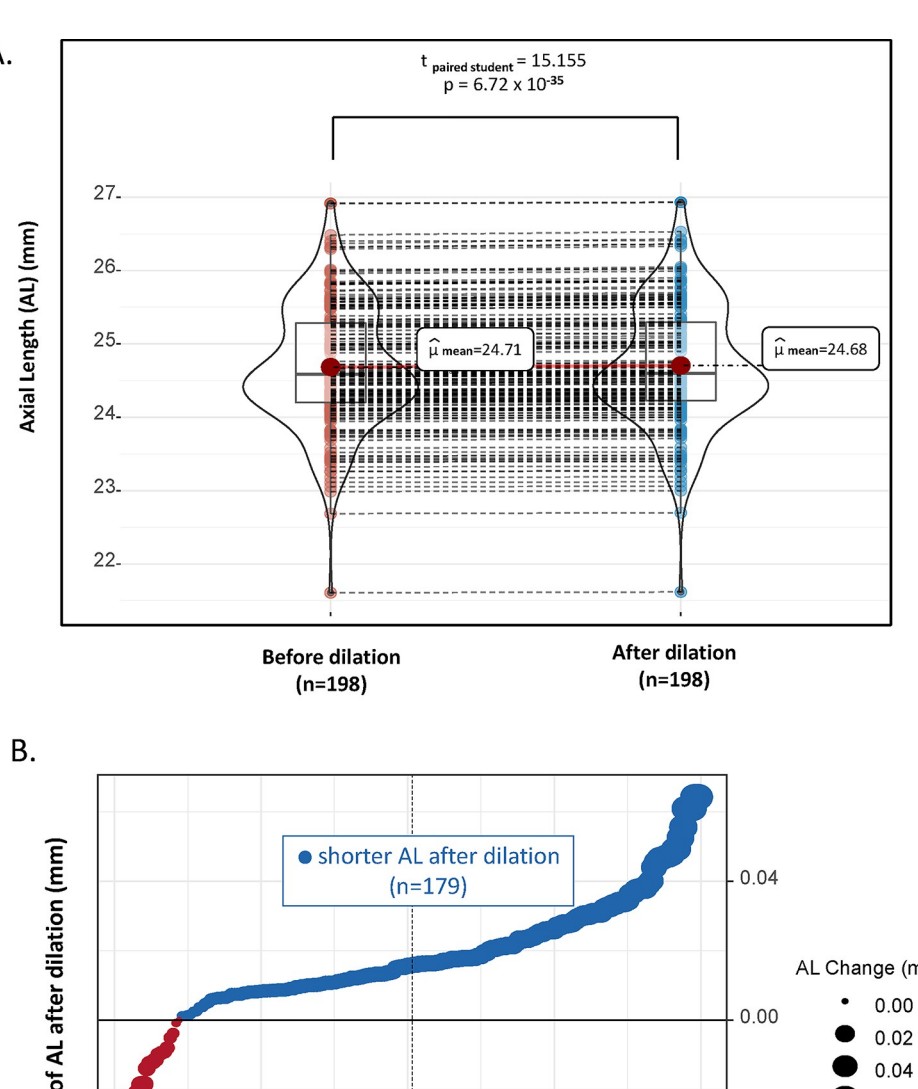

**Fig 1. Change of axial length pre and post pupil dilation.** 1A: The paired violin plot depicts the alteration in axial length before and after dilation. The axial length prior to dilation is longer by 0.028 ± 0.007mm compared to after dilation. In other words, contraction of the ciliary muscle leads to an increase in axial length. 1B: The diagram demonstrates the distribution of cases showcasing a reduction in axial length following dilation. The blue dots symbolize cases in which the axial length diminished after dilation, while the red dots represent cases where the axial length expanded. A vast majority of cases (90.4%) featured a decrease in axial length after pupil dilation.

where a considerable reduction in length was prominent in 90.4% of the study participants. In other words, an average axial length extension of 0.028 ± 0.007mm was observed during ciliary muscle contraction. A finite element model was constructed to demonstrate the transmission of tension from the contraction of such muscle predominantly towards the posterior pole, thereby contributing to an increase in the axial length of the eye.

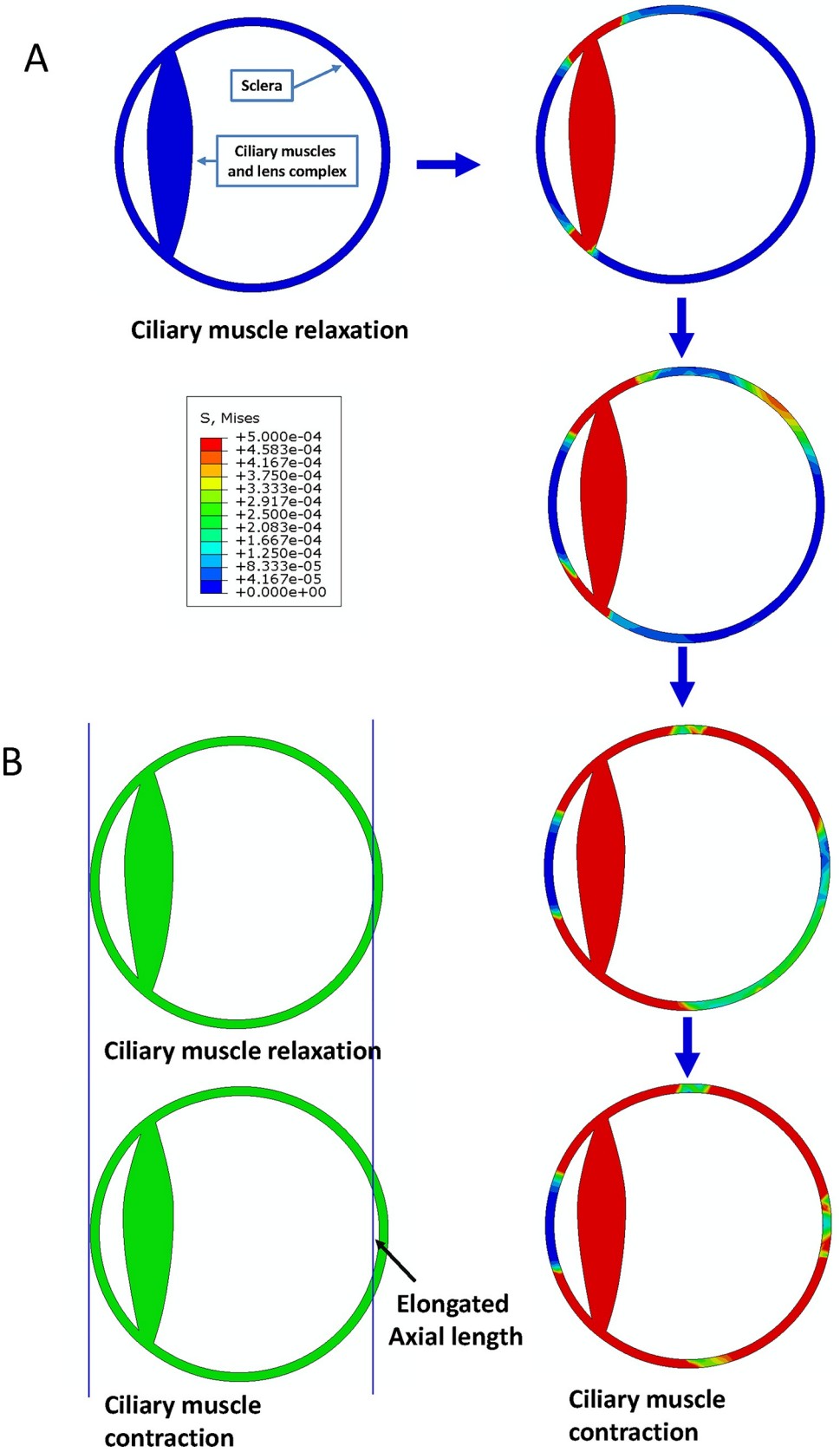

S, Mises

+5.000e-04
+4.583e-04
+4.167e-04
+3.750e-04
+3.333e-04
+2.917e-04
+2.500e-04
+2.083e-04
+1.667e-04
+1.250e-04
+8.333e-05
+4.167e-05
+0.000e+00

**Ciliary muscle relaxation**

**Ciliary muscle relaxation**

**Ciliary muscle contraction**

**Elongated Axial length**

**Ciliary muscle contraction**

**Fig 2. Force distribution of ciliary muscle contraction on the eye ball.** 2A: The finite element model of the eyeball was constructed using ABAQUS software. The ciliary muscle and lens were simplified into a "ciliary muscle-lens complex" The blue color represents the stress condition when the ciliary muscle does not contract. Upon the contraction of the ciliary muscle (change to red color), the tension (depicted in red, yellow, and green) starts from the point of attachment of the ciliary muscle and transmits towards the back of the eye, eventually reaching the posterior pole. 2B: The axial length of the eye extends posteriorly when the ciliary muscles contract.

One previous research has also discovered a shortening of the eye axis after dilation. Bettach et al. assessed the influence of pharmacological pupil dilation on the preciseness of axial length measurement in patients with dense cataracts and previously low-quality axial lengths [20]. They determined that the difference in the axial length before and after pupil dilation was 0.03 ± 0.07 mm. This result corresponds closely with the findings of our research. However, they only analyzed the impact on the accuracy of intraocular lens measurement without linking to the progression of myopia.

Interestingly, the elongation of the eye axis caused by the contraction of the ciliary muscle doesn't appear to be significantly associated with age. Our study concentrated on adolescents averaged 10 years old, contrasting with Bettach et al.'s study which primarily involved senior citizens averaging 73 years old [20]. Despite the significant age difference however, the observed incidence of ocular axis elongation during the ciliary muscle's contraction was remarkably similar in both studies. Kaphle et al. were pioneers in identifying distinct variations in the ciliary muscle dimensions between myopic and emmetropic individuals, observing that those with myopia possess ciliary muscles that are elongated and enlarged compared to their emmetropic counterparts. They found that during the process of accommodation, there was a pronounced thickening of the anterior muscle and a shortening of the curved nasal muscle length, notably more so in myopic eyes than in emmetropic ones. For consistent measurements within the same subject, they advocate the use of a fixed distance approach, while recommending a proportional distance methodology for cross-group refractive comparisons [14].

The biomechanical simulation in this study elucidates the relationship between ciliary muscle contraction and axial growth of the eye. The eye, being spherical, is internally drawn into an ellipsoidal shape when the ciliary muscle contracts. Consequently, this internal reshaping results in an expansion of the eyeball's anteroposterior diameter. It underscores the importance of examining the dynamic interaction between ocular structures for developing comprehensive myopia management approaches.

The importance of this study is highlighted by its potential to return to the "First Principles" in order to elucidate the genesis of myopia, postulating elongation of the eyeball as a possible precipitating factor. Existing research has acknowledged the pivotal role of the transforming growth factor-beta (TGF-beta) in the progression of myopia [17, 21, 22]. However, a lack of comprehensive understanding prevails regarding the activation process of TGF-beta. Previous studies propose that periodical mechanical strain might instigate the activation of TGF-beta [23–25]. TGF-beta activation by traction operates within the broader context of mechano-transduction mechanisms, presenting an intricate process that exposes the pivotal role of physical forces in the regulation of cell function. Cells experience mechanical stimuli through various avenues such as cell-cell interaction, cell-matrix attachment, and exposure to fluid shear stress. These stimuli can induce conformational changes in specific proteins leading to the activation of TGF-beta. As a superfamily of cytokines, TGF-beta plays an essential part in cell proliferation, differentiation, apoptosis, and extracellular matrix production [26]. When operating under mechanical traction, TGF-beta can be activated through the force-induced unfolding of the latency-associated peptide (LAP) which leads to the liberation of active TGF-

beta [27]. This unique activation mechanism implicates traction forces as a critical modulator of TGF-beta signaling pathways and their subsequent biological functions. Elevated levels of TGF-beta may trigger scleral remodeling, promoting the axial elongation characteristic of myopic eyes [28, 29]. Furthermore, these increased levels may also lead to a reduction in collagen synthesis and an increase in matrix metalloproteinases activity, contributing to profound changes in the extracellular matrix [30, 31].

The objective of clarifying the mechanism is to guide the treatment. One significant aspect of this study is its potential to elucidate the mechanisms behind various treatments that decelerate myopia progression. For example, atropine eye drops can induce relaxation in the ciliary muscle, curtailing its pull on the eyeball wall, thereby lowering the probability of ocular elongation [32, 33]. Furthermore, atropine eye drops can increase the eyeball's rigidity, reducing the chances of its elongation [34].

The limitation of this experiment is the inability to track the relationship between ciliary muscle contraction and eye length in real time. We can only observe this relationship pre and post pupil dilation. In the future, our research endeavors will be directed towards the development of methods that facilitate real-time monitoring of the correlation between ciliary muscle contraction and the axial length of the eye. Notably, this study did not examine the interrelationships between the eyeball and the extraocular muscles. Prior researches highlighted the significance of considering the functional entirety and the reciprocal impact of anatomical structures, such as the influence mediated by the Tenon's capsule [35].

In summary, the research hypothesis posited in this study is that the contraction of the ciliary muscle leading to an elongation of the axial length may act as an initiating factor in the onset of myopia. By comparing the axial lengths of the eye under normal ciliary muscle conditions to those when the muscle is in a relaxed state, we observed a slight shortening of the eye's axial length upon muscle relaxation and a slight elongation when the muscle is tense. We further investigated the force distribution of ciliary muscle contraction on the eyeball through computerized mechanical modeling. The findings suggest that ciliary muscle contraction does indeed result in an extension of the axial length, positing it as a potential initiating factor for myopia development.

## Supporting information

**S1 File. The ABAQUS model database of ciliary muscles contraction.** The files included in the attachment can be opened and executed by the ABAQUS 2020 software (Dassault Systèmes SIMULIA Corp.).
(ZIP)

**S1 Fig. The animation of tension changes during ciliary muscle contraction.**
(GIF)

## Author Contributions

**Conceptualization:** Peng Zhou.

**Data curation:** Zhao-Yang Meng, Peng Zhou.

**Methodology:** Peng Zhou.

**Resources:** Lin Yang.

**Software:** Zhao-Yang Meng.

**Supervision:** Peng Zhou.

**Writing – original draft:** Zhao-Yang Meng, Lin Yang, Peng Zhou.

**Writing – review & editing:** Peng Zhou.

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
