## [Decision Letter · Decision Letter 0]

23 Feb 2024

PONE-D-23-39807Ciliary muscles contraction leads to axial length extension —— The possible initiating factor for myopiaPLOS ONE

Dear Dr. Zhou,

Thank you for submitting your manuscript to PLOS ONE. After careful consideration, we feel that it has merit but does not fully meet PLOS ONE’s publication criteria as it currently stands. Therefore, we invite you to submit a revised version of the manuscript that addresses the points raised during the review process.

A learned reviewer have raised several scientific criticisms that need to be addressed while considering revision. Authors should consider revising the manuscript addressing every point raised by the reviewer. 

We look forward to receiving your revised manuscript.

Kind regards,

Sanjoy Bhattacharya

Academic Editor

PLOS ONE

Journal Requirements:

3. We are unable to open your Supporting Information file [Supporting Information 1. The ABAQUS model database of ciliary muscles contraction]. Please kindly revise as necessary and re-upload.

Additional Editor Comments:

A learned reviewer have offered several constructive criticisms. Addressing them satisfactorily will improve the manuscript presentation.

Reviewers' comments:

Reviewer's Responses to Questions

**Comments to the Author**

1. Is the manuscript technically sound, and do the data support the conclusions?

Reviewer #1: Partly

2. Has the statistical analysis been performed appropriately and rigorously? 

Reviewer #1: No

3. Have the authors made all data underlying the findings in their manuscript fully available?

Reviewer #1: Yes

4. Is the manuscript presented in an intelligible fashion and written in standard English?

Reviewer #1: Yes

5. Review Comments to the Author

Reviewer #1: 1. First of all, I suggest referring to the forecast of myopia incidence. This will show the magnitude of the growing problem. DOI: 10.1016/j.ophtha.2016.01.006

2. In the introduction, please find information on the possible mechanism of the Ciliary Muscle and the entire muscular system on the possible formation of myopia. Please refer to the information contained in PMCID paper PMC6969557, where the influence of the Ciliary Muscle on myopia was hypothesised.

3. Information on the connections between the eyeball and extraocular muscles has been shown in papers DOI: 10.3390/jcm12124166 and DOI: 10.1111/opo.12890. I understand that the authors will address the influence of another muscle - the Ciliary Muscle. However, more and more studies emphasise the functional whole and the mutual influence of structures on each other, e.g. through the Tenon's capsule. Therefore, I suggest adding this information and referring to it. This will deepen your understanding of the work.

4. Please also add the research hypothesis at the end of the paper.

5. Regarding the description of methods and materials, please add citations for methodology '' Examination of axial length'', ''Biomechanical simulation''. This will improve the reproducibility of the methodology.

6. Statistical methods are very unclear. Firstly, add sample size calculations. Then add information about the tests used to analyse normality. Then add effect size calculations. A p-value alone is not sufficient - 10.4300/JGME-D-12-00156.1

7. In addition, I see in the results the use of the Pearson test was not stated in the description of the statistic. The current description of the statistic is not reproducible and may therefore be misleading.

8. In addition, always state the result of this test when tesing, which was also missing from the work.

9. The authors use abbreviations such as Ds and Dc. No matter how obvious these abbreviations are, they always need to be developed at the beginning. A note to all authors: if an abbreviation is used for the first time, it should be expanded.

10. ‘’ The biomechanical simulation in this study elucidates the relationship betweeni …. anteroposterior diameter.’’ In this paragraph, develop the concepts I described at the beginning. With reference to the work of PMC6969557,. And to the biomechanical changes in the ocular glabella through the influence on the ocular extraocular muscles (10.3390/jcm12124166, 10.1111/opo.12890)

11. In the discussion, I would also suggest referring more closely to the results of the paper 10.1167/iovs.63.6.24 (paper cited by authors no. 12). This was one of the first papers to note the differences between Ciliary Muscle in Myopia and Emmetropia.

12. I withdraw my assessment of the conclusion until the statistics have been corrected. As I said, the p-value is too open to interpretation at the moment. The key is the effect size.

6. PLOS authors have the option to publish the peer review history of their article (what does this mean?). If published, this will include your full peer review and any attached files.

Reviewer #1: No

---

## [Author Response · Author response to Decision Letter 0]

13 Mar 2024

We would like to extend our profound gratitude for reviewing our manuscript and presenting numerous constructive comments. Following your insightful suggestions, we have amended the manuscript, which has significantly enhanced the quality of our work.

1. First of all, I suggest referring to the forecast of myopia incidence. This will show the magnitude of the growing problem. DOI: 10.1016/j.ophtha.2016.01.006

Response: 

We really appreciated your merit suggestion. We have incorporated the following into our manuscript and have cited the paper: “A systematic review and meta-analysis indicated that the global prevalence of myopia was 22.9%, with 2.7% experiencing high myopia in 2020. Projections suggest a significant escalation by 2050, with an estimated 4758 million individuals (49.8%) likely to be affected by myopia and 938 million (9.8%) by high myopia. (Page 3, Paragraph 1, Line 3) “

2. In the introduction, please find information on the possible mechanism of the Ciliary Muscle and the entire muscular system on the possible formation of myopia. Please refer to the information contained in PMCID paper PMC6969557, where the influence of the Ciliary Muscle on myopia was hypothesised.

Response: 

We really appreciated your suggestion. Indeed, building upon the foundation of previous studies, we have postulated the hypotheses for this research. The following passage, including references, has been incorporated into the manuscript:

“Prior investigations have noted the correlation between ciliary muscle contraction and myopia. Multi-factorial analyses of ocular accommodation, which take into account the morphology of the ciliary muscle, the depth of the anterior chamber, as well as the evaluation of nutritional intake and metabolic indicators, may illuminate the pathogenesis and contribute to the development of innovative management strategies for myopia. Nevertheless, these studies have not delved into the mechanisms by which contractions of the ciliary muscle lead to the elongation of the eye’s axial length. (Page 3, Paragraph 3, Line 7) “

3. Information on the connections between the eyeball and extraocular muscles has been shown in papers DOI: 10.3390/jcm12124166 and DOI: 10.1111/opo.12890. I understand that the authors will address the influence of another muscle - the Ciliary Muscle. However, more and more studies emphasise the functional whole and the mutual influence of structures on each other, e.g. through the Tenon's capsule. Therefore, I suggest adding this information and referring to it. This will deepen your understanding of the work.

Response: 

We really appreciated your suggestion. The manuscript’s discussion section now incorporates the following text, along with the necessary bibliographic references:

“Notably, this study does not examine the interrelationships between the eyeball and the extraocular muscles. Prior research highlights the significance of considering the functional entirety and the reciprocal impact of anatomical structures, such as the influence mediated by the Tenon’s capsule. (Page 13, Paragraph 2, Line 6 ) “

4. Please also add the research hypothesis at the end of the paper.

Response: 

In the discussion section of our manuscript, we have incorporated the following text:

“In summary, the research hypothesis posited in this study is that the contraction of the ciliary muscle leading to an elongation of the axial length may act as an initiating factor in the onset of myopia. By comparing the axial lengths of the eye under normal ciliary muscle conditions to those when the muscle is in a relaxed state, we observed a slight shortening of the eye’s axial length upon muscle relaxation and a slight elongation when the muscle is tense. We further investigated the force distribution of ciliary muscle contraction on the eyeball through computerized mechanical modeling. The findings suggest that ciliary muscle contraction does indeed result in an extension of the axial length, positing it as a potential initiating factor for myopia development. (Page 13, Paragraph 3, Line 1 ) “

5. Regarding the description of methods and materials, please add citations for methodology '' Examination of axial length'', ''Biomechanical simulation''. This will improve the reproducibility of the methodology.

Response: 

We have incorporated the following text, including references, into the Methods section of our manuscript:

“The methodology for axial length measurement is consistent with that used in our previous study. (Page 6, Paragraph 2, Line 1 ) “

“The construction of the Finite Element Model is analogous to that employed in previous studies. (Page 6, Paragraph 4, Line 1 ) “

6. Statistical methods are very unclear. Firstly, add sample size calculations. Then add information about the tests used to analyse normality. Then add effect size calculations. A p-value alone is not sufficient - 10.4300/JGME-D-12-00156.1

Response: 

Thank you very much for your constructive suggestions. We have calculated the sample size as well as the effect size. The actual sample size of this study, with 198 cases, meets the required sample size of 138 cases. The actual difference in the axial length of the eye before and after ciliary muscle relaxation in this study is 0.028, which exceeds the effect size of 0.004. We have added the following content to the Methods section of our paper:

“For the calculation of the required sample size for this study, we employed an online Sample Size Calculator (https://homepage.univie.ac.at/robin.ristl/samplesize.php). The determination of the sample size was informed by preliminary pilot data and guided by relevant literature. The statistical method selected was the paired t-test, with the mean difference set at 0.01 and the standard deviation of differences at 0.03. We established the significance level (alpha) for a two-tailed test at 0.01, and the desired statistical power at 0.9. These parameters yielded a required sample size of 138 pairs. In the present study, a total of 198 eyes were included, fulfilling the criteria for the requisite sample size. 

The data were analyzed using the R programming language (version 4.1.3). Paired student t-test was performed to compare the axial length before dilation (in the contracted state of the ciliary muscle) and after dilation (in the relaxed state of the ciliary muscle). A P-P (Probability-Probability) plot is utilized to ascertain the conformity of a data set with a normal distribution. The Pearson correlation coefficient is employed to assess whether there are differences in the data between both eyes. The significance level was set at p < 0.01.

We employed an online tool to compute the effect size for a before-and-after study (Paired T-test) (https://sample-size.net/effect-size-study-paired-t-test/). The significance level (alpha) for a two-tailed test was set at 0.01, and the desired statistical power at 0.9, with a sample size of 198. Based on the results from a pilot study, the standard deviation of the change in the outcome was established at 0.01627. Subsequent calculations yielded an effect size of 0.004.” (Page 7, Paragraph 2, Line 1 )

7. In addition, I see in the results the use of the Pearson test was not stated in the description of the statistic. The current description of the statistic is not reproducible and may therefore be misleading.

Response: 

We have revised the Statistical Analysis section to include an expanded description of the Pearson test.

8. In addition, always state the result of this test when tesing, which was also missing from the work.

Response: 

We have stated the result of the test when tesing:

“The variation in axial length before and after mydriasis was determined to be statistically significant (t paired student = 15.16, p=6.72 x 10-35). Across both eyes, no apparent axial length differences were identified (Pearson correlation coefficient 96before dilation r(196) =0.9636, P<0.001; after dilation r(196) =0.9507). Considering proportionality, a significant 90.4% (179 out of 198 eyes) exhibited a reduced axial length, while a minor 9.6% (19 out of 198 eyes) demonstrated an increase post-mydriasis.” (Page 9, Paragraph 2, Line 5 ).

9. The authors use abbreviations such as Ds and Dc. No matter how obvious these abbreviations are, they always need to be developed at the beginning. A note to all authors: if an abbreviation is used for the first time, it should be expanded.

Response: 

We are deeply grateful for your suggestions for revision. We have added the following into our manuscript diopter sphere (Ds) (Page 8, Paragraph 2, Line 7 ); diopter of cylinder (DC) (Page 8, Paragraph 2, Line 8 ).

10. ‘’ The biomechanical simulation in this study elucidates the relationship betweeni …. anteroposterior diameter.’’ In this paragraph, develop the concepts I described at the beginning. With reference to the work of PMC6969557,. And to the biomechanical changes in the ocular glabella through the influence on the ocular extraocular muscles (10.3390/jcm12124166, 10.1111/opo.12890)

Response: 

We really appreciated your suggestions. Yes, previous researches have provided academic insights for this study, and this study offers empirical evidence to support the hypotheses established by prior investigations. The following passage, including references, has been incorporated into the manuscript:

Introduction part: “Prior investigations have noted the correlation between ciliary muscle contraction and myopia. Multi-factorial analyses of ocular accommodation, which take into account the morphology of the ciliary muscle, the depth of the anterior chamber, as well as the evaluation of nutritional intake and metabolic indicators, may illuminate the pathogenesis and contribute to the development of innovative management strategies for myopia. Nevertheless, these studies have not delved into the mechanisms by which contractions of the ciliary muscle lead to the elongation of the eye’s axial length. (Page 3, Paragraph 3, Line 7) “

Discussion part: “It underscores the importance of examining the dynamic interaction between ocular structures for developing comprehensive myopia management approaches. (Page 11, Paragraph 3, Line 5 ) “

Discussion part: “Prior researches highlighted the significance of considering the functional entirety and the reciprocal impact of anatomical structures, such as the influence mediated by the Tenon’s capsule.” (Page 13, Paragraph 2, Line 7 )

11. In the discussion, I would also suggest referring more closely to the results of the paper 10.1167/iovs.63.6.24 (paper cited by authors no. 12). This was one of the first papers to note the differences between Ciliary Muscle in Myopia and Emmetropia.

Response: 

Indeed, the discourse surrounding the relationship between the ciliary muscle and myopia is of significant importance. We have incorporated the following content into the discussion section of our article： “Kaphle et al. were pioneers in identifying distinct variations in the ciliary muscle dimensions between myopic and emmetropic individuals, observing that those with myopia possess ciliary muscles that are elongated and enlarged compared to their emmetropic counterparts. They found that during the process of accommodation, there was a pronounced thickening of the anterior muscle and a shortening of the curved nasal muscle length, notably more so in myopic eyes than in emmetropic ones. For consistent measurements within the same subject, they advocate the use of a fixed distance approach, while recommending a proportional distance methodology for cross-group refractive comparisons. “(Page 11, Paragraph 2, Line 7 )

12. I withdraw my assessment of the conclusion until the statistics have been corrected. As I said, the p-value is too open to interpretation at the moment. The key is the effect size.

Response: 

Thank you very much for your valuable suggestions. We have calculated the effect size to be 0.004. Following the relaxation of the ciliary muscle, the axial length of the eye exhibited a decrease of 0.028 ± 0.007 mm. This indicates that the change in axial length is greater than the effect size, rendering the results significant. 

This study corroborates the results found in previous research where relaxation of the ciliary muscle was associated with a shortening of the ocular axis, whereas contraction of the ciliary muscle led to an elongation of the ocular axis. Furthermore, the study employs computational mechanical models to investigate the biomechanical mechanisms by which contraction of the ciliary muscle induces an increase in the ocular axis length. We hop that this research will contribute to the understanding of the pathogenesis of myopia.

---

## [Decision Letter · Decision Letter 1]

25 Mar 2024

Ciliary muscles contraction leads to axial length extension —— The possible initiating factor for myopia

PONE-D-23-39807R1

Dear Dr. Zhou,

We’re pleased to inform you that your manuscript has been judged scientifically suitable for publication and will be formally accepted for publication once it meets all outstanding technical requirements.

Kind regards,

Sanjoy Bhattacharya

Academic Editor

PLOS ONE

Additional Editor Comments (optional):

Reviewers' comments:

Reviewer's Responses to Questions

**Comments to the Author**

1. If the authors have adequately addressed your comments raised in a previous round of review and you feel that this manuscript is now acceptable for publication, you may indicate that here to bypass the “Comments to the Author” section, enter your conflict of interest statement in the “Confidential to Editor” section, and submit your "Accept" recommendation.

Reviewer #1: All comments have been addressed

2. Is the manuscript technically sound, and do the data support the conclusions?

Reviewer #1: (No Response)

3. Has the statistical analysis been performed appropriately and rigorously? 

Reviewer #1: (No Response)

4. Have the authors made all data underlying the findings in their manuscript fully available?

Reviewer #1: (No Response)

5. Is the manuscript presented in an intelligible fashion and written in standard English?

Reviewer #1: (No Response)

6. Review Comments to the Author

Reviewer #1: (No Response)

7. PLOS authors have the option to publish the peer review history of their article (what does this mean?). If published, this will include your full peer review and any attached files.

Reviewer #1: No

---

## [Editor Report · Acceptance letter]

3 Apr 2024

PONE-D-23-39807R1 

PLOS ONE

Dear Dr. Zhou, 

I'm pleased to inform you that your manuscript has been deemed suitable for publication in PLOS ONE. Congratulations! Your manuscript is now being handed over to our production team.

Kind regards, 

on behalf of

Dr. Sanjoy Bhattacharya 

Academic Editor

PLOS ONE